# Impact of COVID-19 on Fetal Outcomes in Pregnant Women: A Systematic Review and Meta-Analysis

**DOI:** 10.3390/jpm13091337

**Published:** 2023-08-30

**Authors:** Rossella Cannarella, Raneen Sawaid Kaiyal, Marta Marino, Sandro La Vignera, Aldo E. Calogero

**Affiliations:** 1Department of Clinical and Experimental Medicine, University of Catania, Via S. Sofia 78, 95123 Catania, Italy; martamarino@outlook.com (M.M.); sandrolavignera@unict.it (S.L.V.); acaloger@unict.it (A.E.C.); 2Glickman Urological and Kidney Institute, Cleveland Clinic Foundation, Cleveland, OH 44195, USA; raneen.sawaid@gmail.com

**Keywords:** COVID-19, pregnancy, miscarriage, birthweight, delivery, SGA, fetal death

## Abstract

**Background:** Coronavirus disease (COVID-19) is a pandemic causing respiratory symptoms, taste alterations, olfactory disturbances, and cutaneous, cardiovascular, and neurological manifestations. Recently, research interest has shifted to reproductive health to understand the factors predisposing to COVID-19 infection in pregnancy, the consequences of the infection on the fetus and on the mother, and possible vertical transmission through the placenta. Pregnancy does not increase the risk of SARS-CoV-2 infection, according to studies. However, contrary to non-pregnant women, pregnancy worsens the clinical outcome of COVID-19. Studies investigating the effects of COVID-19 on pregnancy women are heterogeneous, and the results are often conflicting. **Objectives:** The goal of the current work was to offer a thorough and up-to-date systematic review of, and meta-analysis on, the impact of COVID-19 on ovarian function, pregnancy, and fetal outcomes. **Search strategy:** This meta-analysis (PROSPERO n. CRD42023456904) was conducted using the Preferred Reporting Items for Systematic Review and Meta-Analysis (PRISMA) protocols. The search for relevant material was conducted using PubMed, Scopus, Cochrane, and Embase databases, through to 15 December 2022. **Selection criteria:** Original articles on fertile pregnant women or women attempting to become pregnant, with an active case of, or history of, SARS-CoV-2 infection were included, and reproductive function was compared to that of uninfected women. **Data collection and analysis:** The effects of COVID-19 on female reproductive function, particularly ovarian function, the profile of female sex hormones, pregnancy outcomes and fetal outcomes were the focus of our search. Quantitative analysis was performed with Comprehensive Meta-Analysis Software. The standard difference of the mean was calculated for the statistical comparison between cases and controls. Cochran’s Q test and heterogeneity (I^2^) indexes were used to assess statistical heterogeneity. Sensitivity analysis and publication bias tests were also performed. **Main Results:** Twenty-eight articles met our inclusion criteria, for a total of 27,383 patients pregnant or looking to have offspring, with active or anamnestic COVID-19, and 1,583,772 uninfected control women. Our study revealed that there was no significant difference between COVID-19 patients and the control group in terms of maternal characteristics such as age, body mass index (BMI) and comorbidities that could affect pregnancy and fetal outcomes. The risk of a miscarriage or Cesarean delivery was significantly lower, while the risk of fetal death or premature delivery was significantly higher in COVID-19 patients than in the controls. None of the included studies evaluated hormonal profiles or investigated the presence of infertility. **Conclusions:** Maternal comorbidities, age, and BMI do not raise the risk of COVID-19. However, pregnant women with COVID-19 had a lower risk of miscarriage and Cesarean delivery, possibly because of better prenatal care and high levels of observation during labor. COVID-19 during pregnancy increases the risk of fetal death and premature delivery.

## 1. Introduction

Coronaviruses are a group of viruses that primarily affect humans through zoonotic transmission. They include the Middle East respiratory syndrome coronavirus (MERS-CoV) and the severe acute respiratory syndrome coronavirus (SARS-CoV), which emerged in 2012 and 2003, respectively [1]. In 2019, a novel severe acute respiratory syndrome coronavirus 2 (SARS-CoV-2) etiological agent of a severe acute respiratory syndrome called coronavirus disease (COVID-19) [2] emerged in Wuhan, China. COVID-19 has rapidly become a global pandemic because SARS-CoV-2 has proven to be highly infectious with a high reproductive number (R_0_) and airborne transmission [3].

SARS-CoV-2 has a high predilection for upper and lower respiratory tract cells where it can replicate, causing a variety of symptoms ranging from mild (such as the common cold) to more severe (such as pneumonia and acute respiratory distress syndrome) [4]. In addition to respiratory symptoms, other manifestations are caused by SARS-CoV-2 infection, including taste alterations and smell disorders, skin manifestations, and cardiovascular and neurological manifestations. Furthermore, COVID-19 causes thromboembolic events, as well as a multisystem inflammatory syndrome in children [5].

The disease has also gained attention in obstetrics and gynecology, as pregnant women have weakened immune systems and undergo changes to their cardiopulmonary systems [6]. The prevalence of SARS-CoV-2 infection in pregnant women has varied significantly according to geographical location and country income levels. High rates of 19% [95% confidence interval (CI) 12% to 27%] were reported in Latin America and the Caribbean region, while the lowest rates were reported in East Asia and in the Pacific region (0.4%, 95% CI 0% to 2%) [7].

Concerns about COVID-19 are related to the infection of the mother, placental cells, and the fetus through vertical transmission. Studies have shown that pregnancy does not increase the risk of SARS-CoV-2 infection [8]. However, it exacerbates the clinical outcome of COVID-19 compared to non-pregnant women [9]. Furthermore, pregnant women were more likely to be admitted to an intensive care unit, require invasive ventilation, extracorporeal membrane oxygenation, and die compared to non-pregnant women [6]. Numerous studies have reported that SARS-CoV-2 is capable of infecting placental cells, causing inflammation, infractions, and alterations to maternal and fetal vascular perfusion [10]. Vertical transmission to the fetus, causing intrauterine infection and adverse effects on the conceptus, has also been reported. However, this type of transmission is rare, probably due to the low levels of viremia [11].

On these premises, this study aimed to provide a comprehensive and updated systematic review of, and meta-analysis on, the effects of SARS-CoV-2 infection on ovarian function, pregnancy, and fetal outcomes.

## 2. Material and Methods

### 2.1. Search Strategy

This systematic review and meta-analysis (PROSPERO registration n. CRD42023456904) was carried out according to the Preferred Reporting Items for Systematic Review and Meta-Analysis (PRISMA) protocols. The literature search was performed up to 15 December 2022. An extensive search of PubMed, Scopus, Cochrane, and Embase databases was performed, focusing on the effects of SARS-CoV-2 on female reproductive function, including ovarian function, female sex hormone profile, and pregnancy outcomes.

The key string used was: TITLE-ABS-KEY (“COVID19” OR “COVID-19” OR “SARS-CoV-2” OR “COVID” OR “SARS-CoV” OR “coronavirus” OR “SARS” OR “SARS-CoV”) AND TITLE-ABS-KEY “ovary” OR “estradiol” OR “ovulation” OR “granulosa” OR “oocyte” OR “pregnancy” OR “ART” OR “assisted reproductive tech*” OR “IVF” OR “in vitro fertil*” OR “ICSI” OR “intracytoplasmic sperm injection” OR “IUI” OR “intrauterine insemination” OR “miscarriage” OR “LBR” OR “live birth rate”) AND (LIMIT-TO (DOCTYPE, “ar”)) AND (LIMIT-TO (SUBJAREA, “MEDI”)). Additional manual searches were conducted using the reference lists of relevant studies. No language restriction was applied in any of the literature searches.

### 2.2. Selection Criteria

Articles were assessed for eligibility using the Population, Exposure, Comparison/Comparator, Outcome, and Study Type (PECOS) model system [12]. Specifically, we included articles on fertile pregnant women or women attempting to become pregnant, with an active case or history of SARS-CoV-2 infection and compared their reproductive function to that of uninfected women. Unhealthy women with previously diagnosed reproductive tract diseases (early menopause, hysterectomies, urogenital infections, etc.) that could affect the results were excluded. Original human studies were included, while animal studies, case reports, and non-original studies, such as reviews or comments, were excluded from the analysis. The selection criteria are detailed in Table 1.

The selection of the articles was performed independently by two authors (M.M. and R.K.S). The titles and abstracts of the studies were first independently screened for inclusion. Each decision was reviewed by two non-blinded reviewers (R.C. and S.L.V.). Reviewers resolved disagreements between the authors, while the senior author (A.E.C.) resolved disagreements between the reviewers. Finally, the eligible articles underwent data extraction.

### 2.3. Data Extraction

The following parameters were collected: information on the first author, year of publication, study design, duration of infection (if available), age, body mass index (BMI), maternal comorbidities (if any), LH, follicle-stimulating hormone (FSH), 17β-estradiol (E_2_), progesterone, anti-Müllerian hormone (AMH), anovulation, menstrual irregularities, presence of infertility, pregnancy complications, miscarriages, fetal death, live births, preterm delivery, Cesarean section, gestational age at delivery, small for gestational age (SGA), chorioamnionitis, and birth weight. In particular, data on BMI and maternal comorbidities were extracted to evaluate differences between infected and non-infected women.

When a value was available in a different unit of measure, it was converted according to the conversion tables. For each parameter, the number of women (COVID-19 positive/COVID-19 negative), the mean value, the standard deviation (SD), the median value, and the interquartile range (IQR) range were extracted. For studies expressing data as median and IQR, the formula by Wan and colleagues [13] was used to estimate the mean and SD. Two authors independently extracted the data. Differences between reviewers were discussed until a consensus was reached.

### 2.4. Quality Assessment

The quality of evidence (QoE) in the studies was assessed by R.S.K. In detail, all studies were assessed using the Cambridge Quality Checklists [14]. As no randomized control trials were included, no further evaluation was needed.

### 2.5. Statistical Analysis

Comprehensive Meta-Analysis software (Version 3) (Biostat Inc., Englewood, NJ, USA) was used to analyze the data. Standardized mean difference (SMD) or odd ratio (OR) were used for statistical comparisons between cases and controls. Statistical significance was accepted for *p*-values ≤ 0.05. Statistical heterogeneity was assessed using the heterogeneity index (I^2^) and Cochran’s Q test. In detail, when I^2^ was ≤50%, the variation of the studies was considered homogenous, and the pooled effect size was calculated using the fixed effect model. However, for I^2^ > 50%, indicating significant heterogeneity across the studies, the random effects model was adopted. Finally, the qualitative analysis of publication bias was assessed through the asymmetry of the funnel plot, which suggested some missing studies on one side of the graph. Quantitative analysis of publication bias was instead evaluated using Egger’s intercept test, which indicates the statistical significance of publication bias. In this case, the unbiased estimates were calculated using the “trim and fill” method.

## 3. Results

### 3.1. Study Characteristics

Using the above search strategy, 3552 articles were retrieved. After excluding 84 editorials, book chapters, and reviews, 2894 articles on different topics, 259 articles with unproved SARS-CoV-2 infection, and 199 articles were screened. Of these, 154 were judged to be not pertinent for the purpose of this study after reading their abstracts or full texts. Additionally, 28 studies were excluded as these were animal studies or non-original articles. The remaining 28 studies were carefully read and included in the analysis (Figure 1).

These studies included 27,383 patients with an active or anamnestic COVID-19 infection and 1,583,772 age- and BMI-matched healthy controls with no prior diagnosis of gynecological or pregnancy diseases. The main characteristics of the studies selected for the meta-analysis are described in Table 2.

### 3.2. Quality of Evidence of Included Studies

All the included studies assessed with the Cambridge Quality Checklists scored >6 out of a total of 15. While this scale does not establish a precise threshold for differentiating between high- or low-quality studies, the results suggest that the included studies are of moderate to high methodological quality (Table 3).

### 3.3. Specific Results

The results of the analysis are summarized in Table 4. They include patient and control characteristics, ovarian function, pregnancy outcomes, and fetal outcomes.

#### 3.3.1. Patients and Controls Characteristics

This analysis included age, BMI, and comorbidities (namely, diabetes and hypertension).

*Age:* The pooled analysis of the 12 studies [15,17,23,26,27,29,31,32,33,36,40] included in this meta-analysis revealed that age was not significantly different between the patients and controls (SMD 0.180; CI 95% −0.239; 0.599; *p* = 0.4) (Appendix A). The analysis showed the presence of inter-study heterogeneity, as demonstrated by the I^2^ test (I^2^ = 96%, *p* = 0.000) and the Q-test (Q-value = 293.37). We found no publication bias, as shown by the symmetry of the funnel plots and the result of Egger’s test (intercept 1.40466, 95% CI −6.01106, 8.82039; *p* = 0.34). Consequently, no study was found to be sensitive enough to alter the results (Appendix A).

*BMI:* Four studies [17,24,26,33] reported BMIs for patients and controls, which were not significantly different between the two groups (SMD 0.35; CI 95% −0.178; 0.848; *p* = 0.2) (Appendix A). There was inter-study heterogeneity (I^2^ = 86%, *p* = 0.000; Q = 22.600), as well as an absence of publication bias, as derived from the symmetry of the funnel plots and Egger’s test (intercept 11.95, 95% CI −72.06, 95.96; *p* = 0.30). No study was found to be sensitive enough to alter the results (Appendix A).

*Diabetes:* The analyses of 16 studies [15,16,17,19,24,25,26,28,30,31,33,39,41,44,45] revealed that the risk of diabetes was not significantly different between patients and controls (OR 1.039; CI 95% 0.825; 1.309; *p* = 0.7) (Appendix A), an absence of inter-study heterogeneity, as shown by the I^2^ test (I^2^ = 27%, *p* = 0.128) and the Q-test (Q-value = 21.279), and publication bias, as shown by the symmetry of the funnel plots and the Egger’s test (intercept −0.477, 95% CI −1.62, 0.67; *p* = 0.19). No study was found to be sensitive enough to alter the results (Appendix A).

*Hypertension:* Sixteen studies [15,16,17,19,24,25,26,28,30,31,35,39,41,44,46] were included, showing that the risk of hypertension was not significantly different between patients and controls (OR 1.038; CI 95% 0.796; 1.353; *p* = 0.8) (Appendix A). No inter-study heterogeneity was found, as shown by the I^2^ test (I^2^ = 26.5%, *p* = 0.114) and the Q-test (Q-value = 20.407). Similarly, there was no publication bias, as indicated by the symmetry of the funnel plots and Egger’s test (intercept −0.13, 95% CI −1.53, 1.27; *p* = 0.42). In line with this, no studies were found to be sensitive enough to alter the results (Appendix A).

#### 3.3.2. Ovarian Function

None of the included studies reported data on FSH, LH, E_2_, progesterone, AMH, anovulation, menstrual irregularities, or the presence of infertility. Therefore, this analysis could not be performed.

#### 3.3.3. Pregnancy Outcomes

Data on pregnancy and live birth rates could not be analyzed since they were not reported in the included studies.

*Miscarriages:* Four studies [19,27,37,43] contained information on miscarriages and were included in the analysis. The risk of miscarriage was unexpectedly reduced in patients compared with controls (OR 0.564; CI 95% 0.364; 0.875; *p* = 0.0) (Figure 2).

The analysis showed the absence of inter-study heterogeneity, as indicated by the I^2^ test (I^2^ = 0%, *p* = 0.56, Q-value = 2060), and of publication bias, supported by the symmetry of the funnel plots and Egger’s test (intercept 0.54, 95% CI −2.64, 3.71; *p* = 0.27). However, one study was found to be sensitive enough to alter the results [27]. Indeed, its removal resulted in a non-significant risk of miscarriage in patients compared with controls (OR 0.74; CI 95% 0.25; 2.24; *p* = 0.59) (Appendix A).

*Chorioamnionitis:* A pooled analysis of nine studies [15,20,22,29,30,31,32,33] evaluating the risk for chorioamnionitis did not identify a significantly higher risk in patients compared to that in the controls (OR 0.901; CI 95% 0.579; 1.402; *p* = 0.6) (Figure 3).

The analysis showed the absence of inter-study heterogeneity, as demonstrated by the I^2^ test (I^2^ = 28%, *p* = 0.193, Q-value = 11.154), and publication bias, as indicated by the symmetry of the funnel plots and Egger’s test (intercept −1.02, 95% CI −3.78, 1.75; *p* = 0.21). No study was found to be sensitive enough to alter the results (Appendix A).

#### 3.3.4. Fetal Outcomes

*Birth weight:* A total of nine studies [15,23,26,32,33,36,40,44] were included in the analysis of this outcome, revealing that birth weight was not significantly different between patients and controls (SMD 0.079; CI 95% −0.003; 0.161; *p* = 0.06) (Figure 4), in the absence of inter-study heterogeneity (I^2^ = 34%, *p* = 0.14, Q-value = 12.193), and publication bias, as shown by the funnel plot symmetry and Egger’s test (intercept 0.26, 95% CI −1.78, 2.29; *p* = 0.39). However, three studies were sensitive enough to alter the results [23,26,32].

Accordingly, after their removal, birth weight was significantly higher in patients than in controls (Appendix A).

*Preterm delivery:* A total of 12 studies [16,20,21,23,26,28,30,34,38,41,44,46] evaluated preterm delivery in infected and non-infected women. Their analysis found a significantly higher risk in patients than in controls (OR 2.0; CI 95% 1.827; 2.228; *p* = 0.0) (Figure 5).

No inter-study heterogeneity was found (I^2^ = 47%, *p* = 0.033, Q-value = 21.075), as well as no publication bias, as indicated by the funnel plot symmetry and Egger’s test (intercept −0.29, 95% CI −1.45, 0.86; *p* = 0.29), and no sensitivity studies (Appendix A).

*Cesarean section:* An analysis of 18 studies [15,19,20,21,23,24,25,26,31,32,33,34,36,38,41,46,47] evaluating the prevalence of Cesarean delivery, interestingly, identified a significantly lower risk for patients compared to that of the controls (OR 0.632; CI 95% 0.443; 0.902; *p* = 0.0) (Figure 6).

However, both inter-study heterogeneity (I^2^ = 88%, *p* = 0.000, Q-value = 145,464) and publication bias were found, the latter shown by the funnel plots and Egger’s test (intercept −2.14, 95% CI −3.57, −0.71; *p* = 0.003). No study was found to be sensitive enough to alter the results (Appendix A).

*Gestational age at delivery:* Seven articles [17,19,31,33,36,40,44] reported data on gestational age at delivery, and were therefore included in the analysis; no difference between the patients and controls was found (SMD 0.049; CI 95% −0.147; 0.245; *p* = 0.62) (Figure 7).

However, inter-study heterogeneity was identified by the I^2^ test (I^2^ = 70.7%, *p* = 0.002) and the Q-test (Q-value = 20.509), in the absence of publication bias, as shown by the funnel plot symmetry and Egger’s test (intercept 2.08, 95% CI −1.09, 5.26; *p* = 0.08). In fact, no study was found to be sensitive enough to alter the results (Appendix A).

*Small for gestational age:* An analysis of the eight studies [15,19,21,33,35,39,44] that evaluated the prevalence of children born with SGA in infected and non-infected women found no difference in the risk of SGA between patients and controls (OR 1,0; CI 95% 0.880; 1.149; *p* = 0.9) (Figure 8).

The analysis did not show inter-study heterogeneity, as demonstrated by the I^2^ test (I^2^ = 26%, *p* = 0.219, Q-value = 9497), but did identify the presence of publication bias, as indicated by the funnel plot symmetry and Egger’s test (intercept −0.62, 95% CI −1.93, 0.70; *p* = 0.15). No study was found to be sensitive enough to alter the results (Appendix A).

*Fetal death:* Eight studies [16,18,19,21,23,32,35,36] evaluated fetal death in infected and non-infected women. Their analysis revealed a significantly higher risk in patients than in the controls (OR 1.991; CI 95% 1.783; 2.223; *p* = 0.0) (Figure 9), in the absence of inter-study heterogeneity (I^2^ = 44%, *p* = 0.083, Q-value = 12.589); publication bias, as shown by the funnel plot symmetry, the results of the Egger’s test (intercept −0.62, 95% CI −1.53, 1.27; *p* = 0.18) and the sensitivity studies (Appendix A).

## 4. Discussion

The evolution of a new virus, such as SARS-CoV-2, carries a considerable public health concern, particularly for high-risk groups such as pregnant women. Viral infection during pregnancy puts the mother at risk and has the potential to interfere with pregnancy outcomes and fetus wellbeing, through vertical transmission. During the COVID-19 pandemic, several studies addressed this issue. Therefore, this systematic review and meta-analysis aimed to evaluate the association between COVID-19 infection during pregnancy and pregnancy outcomes. We conducted a thorough literature search and extracted and analyzed the data. Overall, 28 studies were selected and included. We searched for maternal characteristics, including age, BMI, diabetes (both gestational and pre-gestational), hypertension (chronic hypertension and pre-eclampsia), and pregnancy and fetal outcomes.

The results of our study demonstrated that maternal characteristics influencing pregnancy and fetal outcomes were not significantly different between COVID-19 patients and the controls. Therefore, the results of the primary endpoints are not biased by differences in age, BMI, or comorbidities between infected patients and non-infected women. This conclusion is in agreement with what is known so far, namely, that pregnancy or pregnancy-related comorbidities are not risk factors for SARS-CoV-2 infection [43].

Preterm delivery and fetal death were found to be significantly higher in patients with COVID-19. However, the miscarriage rate and the number of Cesarean deliveries were significantly lower. The high prevalence of preterm delivery and fetal death in SARS-CoV-2-infected pregnancies is likely a consequence of the vertical transmission of the virus [48].

Miscarriage and Cesarean section rates were lower in patients with COVID-19, and this could be attributed to some bias in clinical care towards COVID-19 patients, offering them a higher level of care including careful observation, meticulous assistance, and targeted treatment. Regarding the lack of a difference found for birth weight, it should be noted that multiple factors influence this outcome, including parity [49]; comorbidities such as maternal diabetes [50] and hypertension [51]; genetics [52]; and BMI [53]. However, none of the studies conducted multiple regression analyses to evaluate the roles of these factors.

The sensitivity analyses showed that one study [27] for miscarriage and three studies [23,26,32] for birth weight were sensitive enough to skew the results. In the study by Bortoletto and colleagues [27], the cohort included women who became pregnant after the use of assisted reproduction techniques who were compared with historical control patients enrolled one year before the pandemic. The other three studies looked at pregnant women who were asymptomatic or had negative COVID-19 test results. These differences may in part explain why these studies could bias the results. In particular, for birth weight, a lack of symptoms or even negative tests for COVID-19 do not exclude totally the chance of viral infection.

Forty-one meta-analyses on maternal COVID-19 infection and pregnancy outcomes have been published to date. These studies include a high diversity of analyzed data and often reach contradictory conclusions. The number of included studies varies considerably, as well as the types of studies, since some meta-analyses included not only cohort and case–control studies but also case reports [54], case series [55], and brief reports [56]. Some other meta-analyses did not include a control group and the data were presented as prevalence and not as an odd ratio or standard deviation of the mean [55,57,58].

The strengths of our study are several. First, we conducted a systematic literature search nearly two years after the onset of the COVID-19 pandemic. This allowed us to collect a large and comprehensive number of studies. Furthermore, we only included controlled observational studies and assessed their quality, a process which was lacking in most of the previous meta-analyses [54,59]. Our study also includes a sensitivity analysis, contrary to most meta-analyses published so far. Moreover, the analysis of comorbidities included all cases of diabetes in pregnancy (pre-gestational and gestational) and the analysis of hypertension included both chronic hypertension in pregnancy and preeclampsia. We combined these etiologies, as most studies did not include a clear definition of these outcomes. The same limitation applied to low birth weight and Cesarean deliveries. Indeed, most studies did not correctly define low birth weight (below 10th percentile/3rd percentile/less than 2500), and for Cesarean deliveries, there was no difference between urgent Cesarean delivery and elective ones.

## 5. Conclusions

This systematic review and meta-analysis provides a moderate to high level of evidence that maternal age and comorbidities do not increase the risk for SARS-CoV-2 infection. COVID-19 during pregnancy increases the risk of preterm delivery and fetal death. However, pregnant women infected with SARS-CoV-2 had a lower chance of miscarriage and Cesarean delivery, probably due to more careful prenatal care and a higher level of observation during labor.

## Figures and Tables

**Figure 1 jpm-13-01337-f001:**
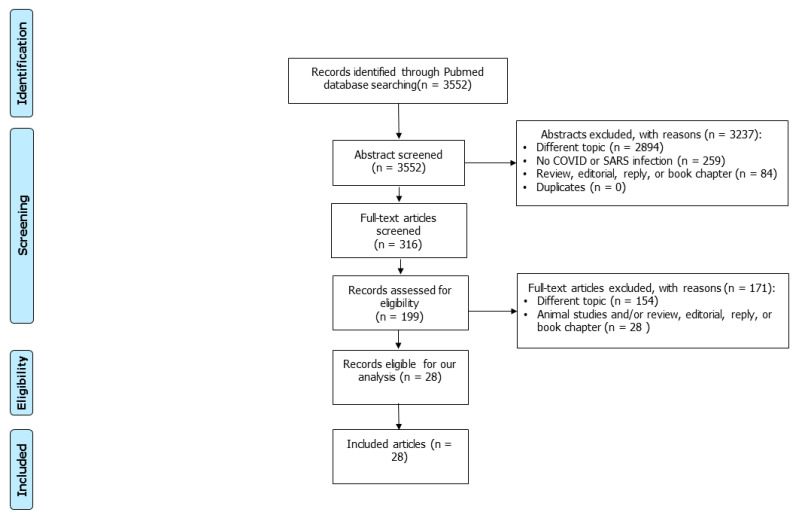
PRISMA flow diagram of the literature screening.

**Figure 2 jpm-13-01337-f002:**
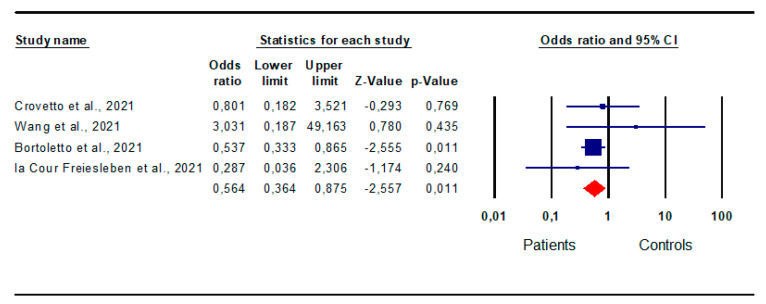
Forest plot of the risk for miscarriage in patients and controls. Crovetto et al., 2021 [19]; Wang et al., 2021 [8]; Bortoletto et al., 2021 [27]; la Cour Freiesleben et al., 2021 [37].

**Figure 3 jpm-13-01337-f003:**
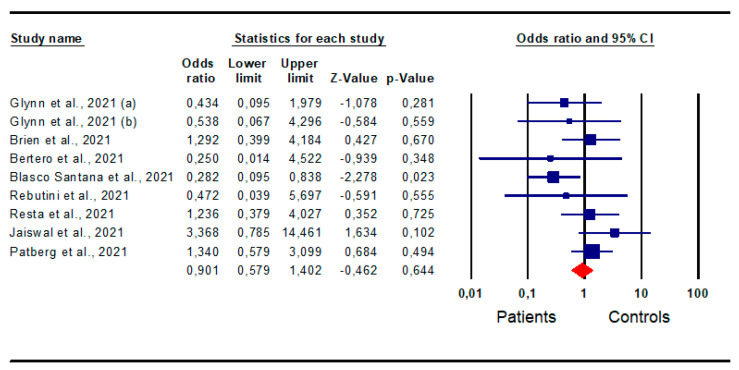
Forest plot of the risk for chorioamnionitis in patients and controls. Glynn et al., 2021 [15]; Brien et al., 2021 [20]; Bertero et al., 2021 [22]; Blasco Santana et al., 2021 [29]; Rebutini et al., 2021 [30]; Resta et al., 2021 [31]; Jaiswal et al., 2021 [32]; Patberg et al., 2021 [33].

**Figure 4 jpm-13-01337-f004:**
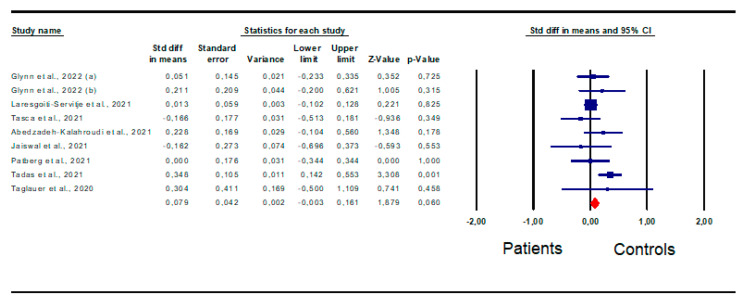
Forest plot of difference in birth weight in patients and controls. Glynn et al., 2022 [15]; Laresgoiti-Servitje et al., 2021 [23]; Tasca et al., 2021 [26]; Abedzadeh-Kalahroudi et al., 2021 [44]; Jaiswal et al., 2021 [32]; Patberg et al., 2021 [33]; Tadas et al., 2021 [36]; Taglauer et al., 2020 [40].

**Figure 5 jpm-13-01337-f005:**
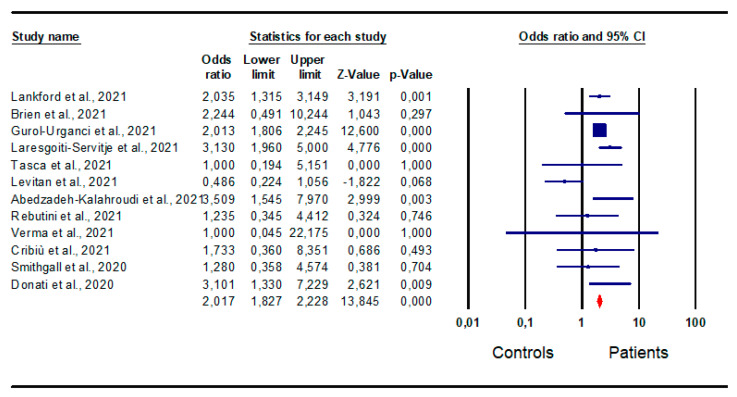
Forest plot of difference in pre-term delivery in patients and controls. Lankford et al., 2021 [16]; Brien et al., [20]; Gurol-Urganci et al., 2021 [21]; Laresgoiti-Servitje et al., 2021 [23]; Tasca et al., 2021 [26]; Levitan et al., 2021 [28]; Abedzadeh-Kalahroudi et al., 2021 [44]; Rebutini et al., 2021 [30]; Verma et al., 2021 [46]; Cribiù et al., 2021 [34]; Smithgall et al., 2020 [38]; Donati et al., 2020 [41].

**Figure 6 jpm-13-01337-f006:**
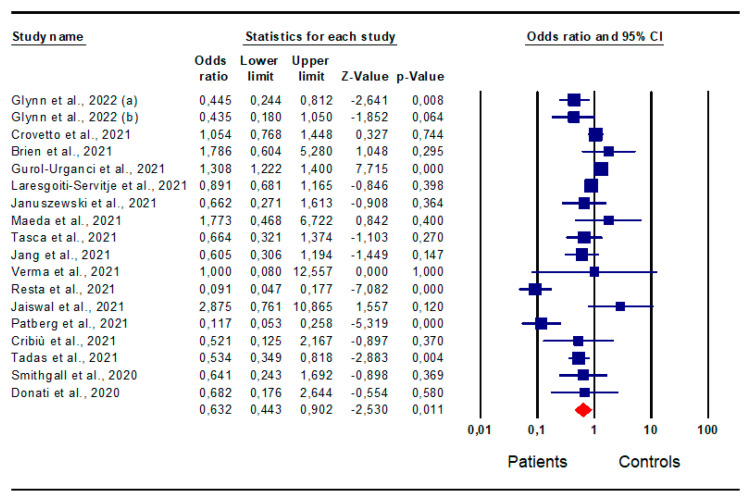
Forest plot of the risk for Cesarean delivery in patients and controls. Glynn et al., 2022 [15]; Crovetto et al., 2021 [19]; Brien et al., [20]; Gurol-Urganci et al., 2021 [21]; Laresgoiti-Servitje et al., 2021 [23]; Januszewski et al., 2021 [24]; Maeda et al., 2021 [25]; Tasca et al., 2021 [26]; Jang et al., 2021 [47]; Verma et al., 2021 [46]; Resta et al., 2021 [31]; Jaiswal et al., 2021 [32]; Patberg et al., 2021 [33]; Cribiù et al., 2021 [34]; Tadas et al., 2021 [36]; Smithgall et al., 2020 [38]; Donati et al., 2020 [41].

**Figure 7 jpm-13-01337-f007:**
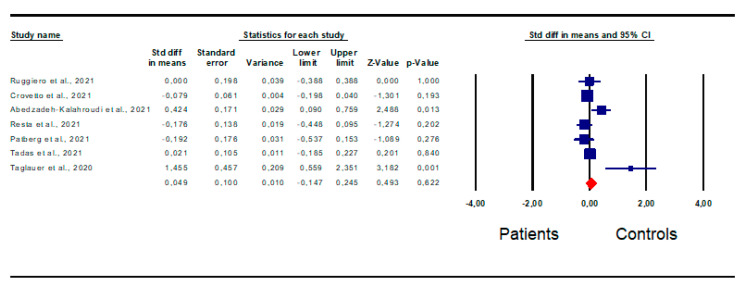
Forest plot of difference in gestational age at delivery in patients and controls. Ruggiero et al., 2021 [17]; Crovetto et al., 2021 [19]; Abedzadeh-Kalahroudi et al., 2021 [44]; Resta et al., 2021 [31]; Patberg et al., 2021 [33]; Tadas et al., 2021 [36]; Taglauer et al., 2020 [40].

**Figure 8 jpm-13-01337-f008:**
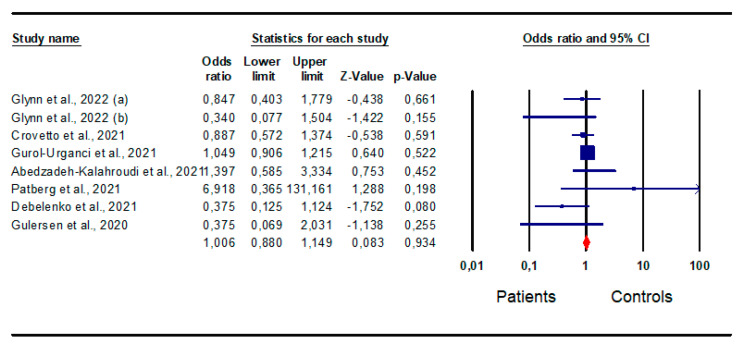
Forest plot of the risk for small for gestational age in patients and controls. Glynn et al., 2022 [15]; Crovetto et al., 2021 [19]; Gurol-Urganci et al., 2021 [21]; Abedzadeh-Kalahroudi et al., 2021 [44]; Patberg et al., 2021 [33]; Debelenko et al., 2021 [35]; Gulersen et al., 2020 [39].

**Figure 9 jpm-13-01337-f009:**
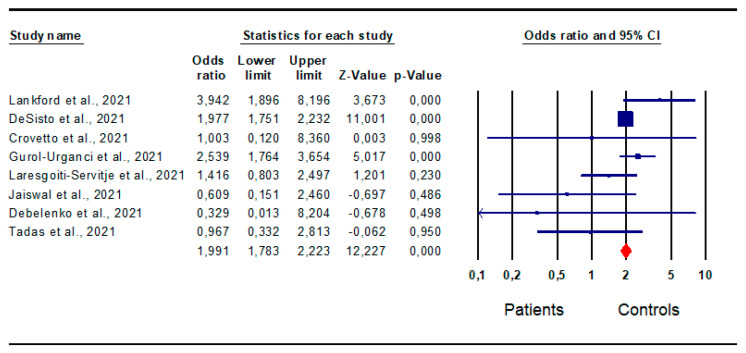
Forest plot of the risk for fetal death in patients and controls. Forest plot of the risk for small for fetal death in patients and controls. Lankford et al., 2021 [16]; DeSisto et al., 2021 [18]; Crovetto et al., 2021 [19]; Gurol-Urganci et al., 2021 [21]; Laresgoiti-Servitje et al., 2021 [23]; Jaiswal et al., 2021 [32]; Debelenko et al., 2021 [35]; Tadas et al., 2021 [36].

**Table 1 jpm-13-01337-t001:** Inclusion and exclusion criteria according to the PECOS model [12].

	Inclusion Criteria	Exclusion Criteria
Population	Fertile pregnant women and/or women seeking pregnancy	Pre-existing comorbidities of the reproductive apparatus (early menopause, hysterectomized woman, urogenital infections, etc.)
Exposure	Previous or current SARS-CoV-2 infection assessed by RT-PCR swab	Presumed or unproven infection
Comparison	No infection	
Outcomes	Ovarian function: ovulation and serum FSH, LH, and E_2_ levels Pregnancy outcomes (both spontaneous and after ART): Pregnancy rate, miscarriage rate, live birth rate, and chorioamnionitis Fetal outcomes: Birthweight, gestational age at delivery, preterm delivery, Cesarean delivery, SGA, and fetal death	
Study type	Observational studies, randomized controlled studies, case–control studies	Animal studies, in vitro studies, reviews, meta-analyses, case reports, book chapters, and editorials

Abbreviations. ART, assisted reproductive technique; FSH, follicle-stimulating hormone; LH, luteinizing hormone; E_2_, 17β-estradiol; RT-PCR, real-time reverse transcription polymerase chain reaction; SARS-CoV-2, severe acute respiratory syndrome coronavirus.

**Table 2 jpm-13-01337-t002:** Characteristics of the 28 studies included in the meta-analysis.

Author, Year	Type of Study	Number of Patients/ Controls	Age (Mean ± SD) (Patients/ Controls)	BMI (Mean ± SD) (Patients/ Controls)	Diabetes (Events/ Total)	Hypertension (Events/ Total)	Gestational Age at Delivery (Years)	Preterm Delivery (Events/ Total)	Fetal Death (Events/ Total)	Live Birth (Events/ Total)
Glyn et al., 2022 [15]	Prospective cohort study	88/188	35.5 ± 5.11/ 35.4 ± 5.3	-	-	-	-	-	-	-
Lankford et al., 2021 [16]	Retrospective cohort study	261/12,046	-	-	Patients:44/261 Controls: 1658/12,046	Patients 15/261 Controls: 454/12,046	-	Patients: 23/261 Controls: 546/12,046	Patients: 8/261 Controls: 96/12,046	Patients: 23/261 Controls: 546/12,046
Ruggiero et al., 2021 [17]	Prospective cohort study	28/287	31.6 ± 7/ 34.2 ± 5	32 ± 7/ 26 ± 4	Patients: 3/28 Controls: 22/287	Patients: 2/28 Controls:6/287	38 ± 1.4/ 39 ± 2	-	-	-
DeSisto et al., 2021 [18]	Retrospective cohort study	21,653/1,227,981	-	-	-	-	-	-	Patients: 273/21,653 Controls: 7881/1,227,981	-
Crovetto et al., 2021 [19]	Prospective cohort study	317/1908	-	-	Patients: 6/317 Controls: 33/1908	Patients: 11/317 Controls: 58/1908	39.1 ± 2.1/ 39.3 ± 2.6	-	Patients: 1/317 Controls: 6/1908	-
Brienet al., 2021 [20]	Prospective cohort study	32/38	-	-	-	-	-	Patients: 5/31 Controls: 3/38	-	-
Gurol-Urganci et al., 2021 [21]	Retrospective cohort study	3527/338,553	-	-	-	-	-	Patients: 369/3527 Controls: 18,527/338,553	Patients: 30/2527 Controls: 1140/338,553	-
Bertero et al., 2021 [22]	Prospective cohort study	18/86	-	-	-	-	-	-	-	-
Laresgoiti-Servitje et al., 2021 [23]	Retrospective cohort study	298/828	28 ± 7.2/ 28 ± 7	-	-	-	-	Patients: 39/298 Controls: 38/828	Patients: 19/298 Controls: 38/828	-
Januszewski et al., 2021 [24]	Retrospective cohort study	47/44	-	30 ± 5/30 ± 5	Patients:10/47 Controls: 7/44	Patients:8/47 Controls: 7/44	-	-	-	-
Wang et al., 2021 [8]	Retrospective cohort study	65/195	-	-	-	-	-	-	-	-
Maeda et al., 2021 [25]	Retrospective cohort study	16/93	-	-	Patients: 2/16 Controls: 2/93	Patients: 0/16 Controls: 14/93	-	-	-	-
Tasca et al., 2021 [26]	Prospective cohort study	64/64	32 ± 5/ 32 ± 6	24 ± 5/ 25 ± 5	Patients: 7/64 Controls: 10/64	Patients: 1/64 Controls: 3/64	-	Patients: 3/64 Controls: 3/64	-	-
Bortoletto et al., 2021 [27]	Cross-sectional survey	202/518	37 ± 4/ 36 ± 4	-	-	-	-	-	-	-
Levitan et al., 2021 [28]	Retrospective case–control study	65/85	-	-	Patients: 2/65 Controls: 11/85	Patients: 9/65 Controls: 13/85	-	Patients: 12/65 Controls: 27/85	-	-
Blasco Santana et al., 2021 [29]	Retrospective cohort study	32/58	32 ± 5.7/ 34 ± 5	-	-	-	-	-	-	-
Rebutini et al., 2021 [30]	Retrospective case–control study.	19/19	-	-	Patients: 4/19 Controls: 3/19	Patients: 3/19 Controls: 3/19	-	Patients: 10/19 Controls: 9/19	-	-
Resta et al., 2021 [31]	Retrospective case–control study.	83/142	33 ± 6.1/ 33 ± 6	-	Patients: 5/83 Controls: 11/142	Patients: 6/83 Controls: 15/142	39 ± 3/ 39 ± 2	-	-	-
Jaiswal et al., 2021 [32]	Prospective cohort study	27/27	27 ± 5/ 25 ± 5	-	-	-	-	-	Patients: 4/27 Controls: 6/27	-
Patberg et al., 2021 [33]	Retrospective cohort study	77/56	29 ± 6/ 32 ± 5	32 ± 6/ 32 ± 5	Patients:7/77 Controls: 0/56	-	39 ± 1/ 39 ± 1	-	-	-
Cribiù et al., 2021 [34]	Prospective cohort study	21/16	-	-	-	-	-	Patients: 6/21 Controls: 3/16	-	-
Debelenko et al., 2021 [35]	Retrospective cohort study	75/75	-	-	Patients: 5/75 Controls: 4/75	Patients: 9/75 Controls: 8/75	-	-	Patients: 0/75 Controls: 1/75	-
Tadas et al., 2021 [36]	Retrospective cohort study	187/181	27 ± 6/ 27 ± 6	-	-	-	38 ± 2/ 38 ± 2	-	Patients: 7/187 Controls: 7/181	-
la Cour Freiesleben et al., 2021 [37]	Retrospective case–control study	18/100	-	-	-	-	-	-	-	-
Smithgall et al., 2021 [38]	Retrospective cohort study	51/25	-	-	-	-	-	Patients: 10/51 Controls: 4/25	-	-
Gulersen et al., 2021 [39]	Retrospective cohort study	50/50	-	-	Patients: 2/50 Controls: 8/50	Patients: 0/50 Controls: 1/50	-	-	-	-
Taglauer et al., 2021 [40]	Prospective cohort study	15/10	32 ± 6/ 30 ± 6	-	-	-	38 ± 6/ 30 ± 6	-	-	-
Donati et al., 2021 [41]	Prospective cohort study	47/99	-	-	Patients: 2/47 Controls: 4/99	Patients: 4/47 Controls: 1/99	-	Patients: 15/47 Controls: 13/99	-	-

**Table 3 jpm-13-01337-t003:** Quality of evidence assessment of the included studies (results of the Cambridge Quality Checklists [14] and Cochrane’s risk-of-bias tool for randomized controlled trials [42]).

Study Name	Type of Study	Cambridge Quality Checklists			
		Checklist for Correlates	Checklist for Risk Factors	Checklist for Causal Risk Factors	Total Score
Glyn et al., 2022 [15]	Prospective cohort study	2	3	5	10
Lankford et al., 2021 [16]	Retrospective cohort study	2	2	5	9
Ruggiero et al., 2021 [17]	Prospective cohort study	2	3	5	10
DeSisto et al., 2021 [18]	Retrospective cohort study	3	2	5	10
Crovetto et al., 2021 [19]	Prospective cohort study	2	3	5	10
Brienet al., 2021 [20]	Prospective cohort study	2	3	5	10
Gurol-Urganci et al., 2021 [21]	Retrospective cohort study	3	2	5	10
Bertero et al., 2021 [22]	Prospective cohort study	2	3	4	9
Laresgoiti-Servitje et al., 2021 [23]	Retrospective cohort study	2	2	5	9
Januszewski et al., 2021 [24]	Retrospective cohort study	3	2	5	10
Wang et al., 2021 [43]	Retrospective cohort study	2	2	5	9
Maeda et al., 2021 [25]	Retrospective cohort study	2	2	5	9
Tasca et al., 2021 [26]	Prospective cohort study	2	3	5	10
Bortoletto et al., 2021 [27]	Cross-sectional survey	2	1	5	8
Levitan et al., 2021 [28]	Retrospective case–control study	2	2	5	9
Blasco Santana et al., 2021 [29]	Retrospective cohort study	2	2	5	9
Rebutini et al., 2021 [30]	Retrospective case–control study	2	2	5	9
Resta et al., 2021 [31]	Retrospective case–control study	3	2	5	10
Jaiswal et al., 2021 [32]	Prospective cohort study	3	3	5	11
Patberg et al., 2021 [33]	Retrospective cohort study	3	2	5	10
Cribiù et al., 2021 [34]	Prospective cohort study	2	3	5	10
Debelenko et al., 2021 [35]	Retrospective cohort study	2	2	5	9
Tadas et al., 2021 [36]	Retrospective cohort study	2	2	5	9
la Cour Freiesleben et al., 2021 [37]	Retrospective case–control study	2	2	5	9
Smithgall et al., 2021 [38]	Retrospective cohort study	2	2	5	9
Gulersen et al., 2021 [39]	Retrospective cohort study	2	2	5	9
Taglauer et al., 2021 [40]	Prospective cohort study	3	3	5	11
Donati et al., 2021 [41]	Prospective cohort study	2	3	5	10

**Table 4 jpm-13-01337-t004:** Summary of the results of the present meta-analysis.

Parameters	N° of Studies	Patients Included	SD in Means	CI 95%	OR	*p*-Value	Interpretation
Age	12	1101	0.180	−0.239; 0.599		0.4	Non-significantly different
BMI	4	216	0.335	−0.178; 0.848		0.2	Non-significantly different
Risk of diabetes	16	1173		0.825; 1.309	1.039	0.7	Non-significantly different
Risk of hypertension	16	1160		0.796; 1.353	1.038	0.8	Non-significantly different
Miscarriages	4	602		0.364; 0.875	0.564	0.0	Significantly lower in patients
Chorioamnionitis	9	376		0.579; 1.402	0.901	0.6	Non-significantly different
Birthweight	9	756	0.079	−0.003; 0.161		0.06	Non-significantly different
Small for gestational age	8	4134		0.880; 1.149	1.0	0.9	Non-significantly different
Gestational age at delivery	7	707	0.049	−0.147; 0.245		0.62	Non-significantly different
Pre-term delivery	12	4299		1.827; 2.228	2.017	0.0	Significantly higher in patients
Cesarean delivery	18	4882		0.443; 0.902	0.632	0.0	Significantly lower in patients
Risk of fetal death	8	26,345		1.783; 2.223	1.991	0.0	Significantly higher in patients

Abbreviations: BMI, body mass index; SD, standard difference; CI, confidence interval; OR, odds ratio.

## Data Availability

No original data were generated in this study.

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
