# Peer review of "Impact of COVID-19 on Fetal Outcomes in Pregnant Women: A Systematic Review and Meta-Analysis"

_jpm, 2023, doi:10.3390/jpm13091337_

Round 1

Reviewer 1 Report

Overall Comments:

The authors present a Systematic Review and Meta-Analysis examining the impact of COVID-19 on ovarian function, pregnancy, and fetal outcomes.

The authors found that maternal comorbidities, age, and body mass index do not raise the risk of COVID-19. However, pregnant women with COVID-19 had a lower chance of miscarriage and cesarean delivery, possibly because of better prenatal care and high levels of observation during labor. COVID-19 during pregnancy increases the risk of fetal death and premature delivery.

I found the study very carefully designed and developed, the question posed by the authors is well defined on page 4 (lines 72-80): “Concerns about CoVID-19 are related to the infection of the mother, placental cells, and the fetus through vertical transmission. Studies have shown that pregnancy does not increase the risk of SARS-CoV-2 infection. However, it exacerbates the clinical outcome of COVID-19 compared to non-pregnant women. Furthermore, pregnant women were more likely to be admitted to an intensive care unit, require invasive ventilation, extracorporeal membrane oxygenation, and die compared to non-pregnant women. Numerous studies have reported that SARS-CoV2 is capable of infecting placental cells causing inflammation, infractions, and alterations in maternal and fetal vascular perfusion. Vertical transmission to the fetus causing intrauterine infection and adverse effects on the conceptus has also been reported.”

The methods are appropriate and very interesting from the methodological-scientific point of view, an intensive data collection work has been made by the authors.

The discussion and conclusions are well-balanced and adequately supported by the data.

I found that from the research point of view, the study has merit and a high scientific level. 

ACCEPTABLE WITH MINOR MODIFICATIONS

Specific Comment:

Please register the study in PROSPERO, it is very important.

Page 5, line 101. PECOS? Please correct.

While in the text you differentiate between patients and controls, in the figures it says A and B, please clarify at the beginning of the Results Section what A and B correspond to. Otherwise, it confuses the reader.

For Odd ratio analysis, please specify (if possible) whether a fixed or random model was used.

Please provide the PRISMA Checklist (attached).

Author Response

Answers to the Reviewer #1 comments

Manuscript ID jpm-2571117

Comment 1. The authors present a Systematic Review and Meta-Analysis examining the impact of COVID-19 on ovarian function, pregnancy, and fetal outcomes. The authors found that maternal comorbidities, age, and body mass index do not raise the risk of COVID-19. However, pregnant women with COVID-19 had a lower chance of miscarriage and cesarean delivery, possibly because of better prenatal care and high levels of observation during labor. COVID-19 during pregnancy increases the risk of fetal death and premature delivery. I found the study very carefully designed and developed, the question posed by the authors is well defined on page 4 (lines 72-80): “Concerns about CoVID-19 are related to the infection of the mother, placental cells, and the fetus through vertical transmission. Studies have shown that pregnancy does not increase the risk of SARS-CoV-2 infection. However, it exacerbates the clinical outcome of COVID-19 compared to non-pregnant women. Furthermore, pregnant women were more likely to be admitted to an intensive care unit, require invasive ventilation, extracorporeal membrane oxygenation, and die compared to non-pregnant women. Numerous studies have reported that SARS-CoV2 is capable of infecting placental cells causing inflammation, infractions, and alterations in maternal and fetal vascular perfusion. Vertical transmission to the fetus causing intrauterine infection and adverse effects on the conceptus has also been reported.” The methods are appropriate and very interesting from the methodological-scientific point of view, an intensive data collection work has been made by the authors. The discussion and conclusions are well-balanced and adequately supported by the data. I found that from the research point of view, the study has merit and a high scientific level. ACCEPTABLE WITH MINOR MODIFICATIONS.

Answer to comment 1. We sincerely thank the Reviewer for the words of appreciation for our study. We have carefully reviewed our manuscript based on the comments raised and the changes made are shown below.

Comment 2. Please register the study in PROSPERO, it is very important.

Answer to comment 2. Thank you for this suggestion. We have now registered the study in PROSPERO. Its number is CRD42023456904 (please see lines 33 and 89).

Comment 3. Page 5, line 101. PECOS? Please correct.

Answer to comment 3. It is PECOS and not PICOS because this is not an intervention review but an exposure review. In other words, we assessed exposure to an agent (CoVID-19) and not an intervention, in patients versus controls. Therefore, the acronym PECOS is appropriate.

Comment 4. While in the text you differentiate between patients and controls, in the figures it says A and B, please clarify at the beginning of the Results Section what A and B correspond to. Otherwise, it confuses the reader.

Answer to comment 4. Thank you for this comment. We have changed all the figures, replacing A and B with Patients and Controls.

Comment 5. For Odd ratio analysis, please specify (if possible) whether a fixed or random model was used.

Answer to comment 5. For both SMD and OR, we used the fixed model for I2 ≤50% and the random model for I2 >50% (please see lines 142-148).

Comment 6. Please provide the PRISMA Checklist (attached).

Answer to comment 6. Done, as requested. Thank you.

Reviewer 2 Report

Reviewer Comments

Title

It seems that the objective of the authors is to identify whether the history of COVID-19 infection has any impact on fertility, in addition to the effects on the fetus and the pregnant woman, but they do not include in the title anything about fertility, ovarian function or female sex hormones.

Introduction

During the development of this section, the authors do not describe the effects or repercussions of the COVID-19 infection on fertility or ovarian hormonal physiology, despite mentioning it as the objective of this work.

Materials and Methods

Within the authors' bibliographic search, the number of references related to fertility problems and ovarian hormonal function is minimal.

Results

There are no results on fertility problems and a history of COVID-19 infection, as well as no data on ovarian function.

Discussion

The authors only develop the results of other systematic reviews and meta-analyses, they do not include results on ovarian function.

Conclusion

It is similar to those described in other publications on systematic reviews and meta-analyses.

Additional comments

Purpose of the document

The objective of the research work was to demonstrate the impact of the COVID-19 infection on pregnant women, the fetus and fertility; however the authors do not develop anything in the manuscript about the impact on fertility or ovarian hormonal function.

What is the main question addressed by the research?

The main research question raised with this manuscript is the impact of COVID-19 infection on perinatal outcomes and fertility, however nothing is developed on the impact on fertility.

Do you consider the topic original or relevant in the field?

The subject of the possible repercussions on fertility is of the utmost importance, but it does not contain information about it.

Do you address a specific gap in the field?

Considering the number of couples that have fertility problems, knowing whether the history of COVID-19 infection represents an extra factor to consider within the study of the infertile couple would have been very useful.

Main contributions

What does it add to the subject area compared to other published material?

This work does not add information to the published systematic reviews and meta-analyses on the perinatal impact of COVID-19 infection.

Findings

What specific improvements should the authors consider regarding the methodology?

Increase the bibliographic search on the repercussions on fertility due to a history of COVID-19 infection.

What additional controls should be considered?

If the authors wish to demonstrate that COID-19 infection has any impact on fertility, they should consider hormonal determinations of ovarian function after COVID-19 infection in pregnancy and in women without COVID-19 infection.

Are the conclusions consistent with the evidence and arguments presented and do they address the main question posed?

The conclusion is similar to that described in other systematic reviews and meta-analyses.

Issue you think needs to be addressed

If the authors are interested in the impact of a history of COVID-19 infection on fertility, seek information regarding the behavior of ovarian function in women who had COVID-19 infection during pregnancy.

Author Response

Answers to the Reviewer #2 comments

Manuscript ID jpm-2571117

Comment 1. Title: It seems that the objective of the authors is to identify whether the history of COVID-19 infection has any impact on fertility, in addition to the effects on the fetus and the pregnant woman, but they do not include in the title anything about fertility, ovarian function or female sex hormones.

Answer to comment 1. Thank you for this comment. As reported in the study objective and also in the PECOS table, this systematic review and meta-analysis aimed to analyze the following endpoints in fertile pregnant women and/or women seeking pregnancy with previous or current SARS-CoV2 infection assessed by RT-PCR swab versus uninfected controls from the same population:

  • Ovarian function: ovulation and serum levels of FSH, LH, and E2
  • Pregnancy outcomes (both spontaneous and after ART): Pregnancy rate, miscarriage rate, live birth rate, and chorioamnionitis
  • Fetal outcomes: Birthweight, gestational age at delivery, preterm delivery, cesarean delivery, SGA, and fetal death

As explained in the result section (please see 3.3.2 Ovarian function), “None of the included studies reported data on FSH, LH, E2, progesterone, AMH, anovulation, menstrual irregularities, or the presence of infertility. Therefore, this analysis could not be performed”. In particular, the screened studies including information on ovarian function were descriptive, with no control groups, and, therefore, were not suitable for our meta-analysis. This is the reason why the title does not include information on ovarian function.

Comment 2. Introduction: During the development of this section, the authors do not describe the effects or repercussions of the COVID-19 infection on fertility or ovarian hormonal physiology, despite mentioning it as the objective of this work.

Answer to comment 2. Please see the “Answer to comment 1”. This information is not reported in the Introduction as none of the articles that met our inclusion criteria included the ovarian function as an endpoint measure.

Comment 3. Materials and Methods: Within the authors' bibliographic search, the number of references related to fertility problems and ovarian hormonal function is minimal.

Answer to comment 3. Please see the “Answer to comment 1”. We meticulously screened a huge number of articles (n=3,552). The screened studies that included information on ovarian function were descriptive and without control groups. For this reason, they were not suitable for our meta-analysis,

Comment 4. Results: There are no results on fertility problems and a history of COVID-19 infection, as well as no data on ovarian function.

Answer to comment 4. Please see the “Answer to comment 1” and the Section 3.3.2.

Comment 5. Discussion: The authors only develop the results of other systematic reviews and meta-analyses, they do not include results on ovarian function.

Answer to comment 5. Please see the “Answer to comment 1”.

Comment 6. Conclusion: It is similar to those described in other publications on systematic reviews and meta-analyses.

Answer to comment 6. We disagree with this statement. Previously published meta-analyses often come to contradictory conclusions. They analyze a high diversity of data, and statistical methods do not always include sensitivity analysis or publication bias analysis. Furthermore, the inclusion criteria are not always the same. Please see lines 329-330.

Comment 7. Purpose of the document: The objective of the research work was to demonstrate the impact of the COVID-19 infection on pregnant women, the fetus and fertility; however the authors do not develop anything in the manuscript about the impact on fertility or ovarian hormonal function.

Answer to comment 7. Please see the “Answer to comment 1”.

Comment 8. What is the main question addressed by the research? The main research question raised with this manuscript is the impact of COVID-19 infection on perinatal outcomes and fertility, however nothing is developed on the impact on fertility.

Answer to comment 8. Please see the “Answer to comment 1”.

Comment 9. Do you consider the topic original or relevant in the field? The subject of the possible repercussions on fertility is of the utmost importance, but it does not contain information about it.

Answer to comment 9. Please see the “Answer to comment 1”.

Comment 10. Do you address a specific gap in the field? Considering the number of couples that have fertility problems, knowing whether the history of COVID-19 infection represents an extra factor to consider within the study of the infertile couple would have been very useful.

Answer to comment 10. Please see the “Answer to comment 1”.

Comment 11. Main contributions: What does it add to the subject area compared to other published material? This work does not add information to the published systematic reviews and meta-analyses on the perinatal impact of COVID-19 infection.

Answer to comment 11. Please see the “Answer to comment 6”.

Comment 12. Findings: What specific improvements should the authors consider regarding the methodology? Increase the bibliographic search on the repercussions on fertility due to a history of COVID-19 infection.

Answer to comment 12. Please see the “Answer to comment 1”.

Comment 13. What additional controls should be considered? If the authors wish to demonstrate that COID-19 infection has any impact on fertility, they should consider hormonal determinations of ovarian function after COVID-19 infection in pregnancy and in women without COVID-19 infection.

Answer to comment 13. Please see the “Answer to comment 1”.

Comment 14. Are the conclusions consistent with the evidence and arguments presented and do they address the main question posed? The conclusion is similar to that described in other systematic reviews and meta-analyses.

Answer to comment 14. Please see the “Answer to comment 6”.

Comment 15. Issue you think needs to be addressed. If the authors are interested in the impact of a history of COVID-19 infection on fertility, seek information regarding the behavior of ovarian function in women who had COVID-19 infection during pregnancy.

Answer to comment 15. Please see the “Answer to comment 1”.